# Experimental Autoimmune Encephalomyelitis of Mice: Enzymatic Cross Site-Specific Hydrolysis of H4 Histone by IgGs against Histones and Myelin Basic Protein

**DOI:** 10.3390/ijms23169182

**Published:** 2022-08-16

**Authors:** Andrey E. Urusov, Kseniya S. Aulova, Pavel S. Dmitrenok, Valentina N. Buneva, Georgy A. Nevinsky

**Affiliations:** 1Institute of Chemical Biology and Fundamental Medicine of the Siberian Division of Russian Academy of Sciences, Lavrentiev Ave. 8, 630090 Novosibirsk, Russia; 2G. B. Elyakov Pacific Institute of Bioorganic Chemistry, Far East Division, Russian Academy of Sciences, 690022 Vladivostok, Russia

**Keywords:** C57BL/6 mice, EAE model of human multiple sclerosis, immunization mice with MOG and DNA–histones complex, catalytic antibodies, hydrolysis of histones and myelin basic protein, cross-complexation and catalytic cross-reactivity

## Abstract

Histones play vital roles in chromatin functioning and gene transcription, but in intercellular space, they are harmful due to stimulating systemic inflammatory and toxic responses. Myelin basic protein (MBP) is the most important protein of the axon myelin–proteolipid sheath. Antibodies-abzymes with different catalytic activities are critical and specific features of some autoimmune diseases. Five IgG preparations against histones (H4, H1, H2A, H2B, and H3) and against MBP corresponding to different spontaneous, MOG (myelin oligodendrocyte glycoprotein of mice), and DNA–histones that accelerated onset, acute, and remission stages of experimental autoimmune encephalomyelitis (EAE; model of human multiple sclerosis) development were obtained from EAE-prone C57BL/6 mice by several affinity chromatographies. IgG-abzymes against five histones and MBP possess unusual polyreactivity in complexation and catalytic cross-reactivity in the hydrolysis of histone H4. IgGs against five histones and MBP corresponding to 3 month-old mice (zero time) in comparison with Abs corresponding to spontaneous development of EAE during 60 days differ in type and number of H4 sites for hydrolysis. Immunization of mice with MOG and DNA–histones complex results in an acceleration of EAE development associated with an increase in the activity of antibodies in H4 hydrolysis. Twenty days after mouse immunization with MOG or DNA–histones complex, the IgGs hydrolyze H4 at other additional sites compared to zero time. The maximum number of different sites of H4 hydrolysis was revealed for IgGs against five histones and MBP at 60 days after immunization of mice with MOG and DNA–histones. Overall, it first showed that at different stages of EAE development, abzymes could significantly differ in specific sites of H4 hydrolysis.

## 1. Introduction

Histones and their modified forms play a vital role in chromatin functioning. Free extracellular histones usually act as damage molecules [1]. Treatment of experimental animals with exogenous histones leads to systemic toxic responses due to inflammatory reactions and Toll-like receptor activation [1]. Animal treatment by anti-histone neutralizing antibodies (Abs), heparin, thrombomodulin, and activated protein C leads to the protection of mice against ischemia-reperfusion injury, sepsis, lethal endotoxemia, trauma, peritonitis, stroke, pancreatitis, coagulation, and thrombosis. Moreover, the increase in the free histones and nucleosome fragments in the blood results in several pathophysiological processes, including progression in inflammatory, several autoimmune diseases (ADs), and cancer [1]. 

A central tetramer in the core of nucleosome particles contains two molecules of H4 and H3 circled by two dimers of histones H2B and H2A linked to two supercoiled turns of double-stranded (ds) DNA [2]. H1 participates in packing chromatin substructures into a higher-order structure, and H1 is first displaced from chromatin when treated with alkali or acid. The H3 and H4 **histones** demonstrate the maximum transfection efficiency of all tested agents [3]. 

In systemic lupus erythematosus (SLE), multiple sclerosis (MS), and some other autoimmune diseases, autoantibodies (auto-Abs) against DNA are directed against nucleosomal histone–DNA complexes emerging in the blood due to cell apoptosis [4].

Multiple sclerosis (MS) is an inflammatory-demyelinating pathology of the central nervous system (CNS). It is characterized by perivascular infiltrates containing largely macrophages and T lymphocytes [5]. The activated myelin-reactive CD4^+^ T cells might be the principal mediators of MS [5]. Some data specify a vital role of B cells and auto-Abs against myelin autoantigens, including myelin basic protein (MBP) in MS pathogenesis [5,6,7].

There are several various experimental autoimmune encephalomyelitis (EAE) mouse models that mimic a specific facet of multiple sclerosis well (for review, see [8,9]). Autoimmune diseases were first suspected to originate from bone marrow hematopoietic stem cells (HSCs) defects [10]. It was proved later that the spontaneous and antigen-induced development of ADs is reached due to specific immune reorganization of bone marrow HSCs [11,12,13,14,15,16,17]. EAE-prone C57BL/6 mice were used recently for the study of possible mechanisms of spontaneous, DNA–protein complexes [11,12], and myelin oligodendrocyte glycoprotein (MOG)-accelerated development of EAE [13,14]. It was demonstrated that immunization of systemic lupus erythematosus (SLE)-prone MRL-lpr/lpr mice with DNA–protein complexes [15,16,17] as well as C57BL/6 mice with DNA–histone complexes and MOG [11,12,13,14] results in an acceleration of SLE and EAE development. The acceleration is associated with specific changes in HSC differentiation profiles, an increase in lymphocyte proliferation, and repression of apoptosis in different organs of mice [11,12,13,14,15,16,17]. These changes in parallel are associated with the production of autoantibodies-abzymes (Abzs) splitting DNA, RNA, polysaccharides, proteins, and peptides. The detection of different antibodies-abzymes is the most statistically significant and earliest marker of many ADs [11,12,13,14,15,16,17,18,19,20,21,22,23]. Catalytic activities of Abzs are well detected even at the very initial stages of the autoimmune (AI) pathologies (at the pre-disease onset stage) before the disclosure of typical markers of different ADs [13,14,15,16,17]. 

Titers of auto-Abs to specific autoantigens at the beginning of different ADs usually correspond to typical index ranges conforming to healthy humans and mice. The appearance of multiple abzymes clearly indicates the start of autoimmune pathologies, while an increment in their catalytic activities is associated with profound pathology development. However, several parallel mechanisms might provide different AD development, eventually leading to a self-tolerance breakdown [20,21,22,23].

Natural Abzs splitting different peptides, proteins, oligosaccharides, DNA, and RNA, were detected in sera of patients with several ADs and viral diseases [18,19,20,21,22,23,24,25,26,27]. 

Abzymes with shallow activities hydrolyzing polysaccharides [28], thyroglobulin [29], and vasoactive neuropeptide [30] were discovered in sera of some healthy volunteers. Healthy humans, however, usually lack abzymes [18,19,20,21,22,23]. However, some germline auto-Abs of healthy people can possess high levels of superantigen-, amyloid-, and microbe-directed activities [31,32].

Similar to SLE patients [23], the blood sera of MS patients contain abzymes hydrolyzing DNA and RNA [33,34,35], MBP [36,37,38,39], oligosaccharides [19], and histones [40]. Relative activities of IgG-abzymes from the cerebrospinal fluids of MS patients splitting polysaccharides, MBP, and DNA are on average from 30 to 60 times higher than those isolated from the blood sera of the same patients [41,42,43]. In various ADs, Abzs splitting MBP can attack this protein in the myelin-proteolipid sheath of axons and, therefore, could play a very harmful role in the pathogenesis of MS, SLE, and other AI diseases [19,20,21,22,23].

Abzymes hydrolyzing five histones (H4, H3, H2A, H2B, and H1) are detected in sera of MS [37], HIV-infected patients [44,45,46,47,48,49], and EAE mice [11]. As mentioned above, free extracellular histones act as damage molecules [1]. Complexes of DNA with histones are known as the most critical antigens in producing auto-Abs against DNA and histones [4], which are very hazardous for mammals. They can penetrate through cellular and nuclear membranes, split chromatin DNA, and induce cell apoptosis [50,51,52]. Therefore, abzymes that hydrolyze MBP, DNA, and histones may be involved in the pathogenesis of MS and other ADs.

It is believed that the development of various ADs could be associated with human infection by different viruses and/or bacteria (human herpesvirus, human endogenous retroviruses, and Epstein–Barr virus) (for a review, see [53,54,55,56]). At first, there could be the synthesis of antibodies against viral or bacterial compounds, which may be structurally similar to human proteins [57,58]. Then, due to the mimicry of any viral or bacterial protein with those of human ones, there may be immune system violation resulting in the generation of autoantibodies to human substances and the development of ADs. In addition, the immunization of different autoimmune-prone mice leads to a significantly higher incidence of abzymes synthesis with higher catalytic activities than in normal conventionally used mouse strains [59,60]. 

Unspecific complex formation of different enzymes with foreign ligands is a widespread phenomenon [61,62,63]. The efficiency of correct substrate selection by enzymes on the stage of complexation is usually only 1–2 orders of magnitude [61,62,63]. It is subsequent changes in substrate confirmation at the stage of the catalysis that directly increases the reaction rate by 5–8 orders of magnitude for specific compared with the non-specific substrates [61,62,63]. Therefore, catalytic cross-reactivity in the case of normal-classical enzymes is a sporadic case [61,62,63]. Typically, normal enzymes catalyze only one chemical reaction.

Non-specific complexation of some proteins with Abs against other ones discovered by affinity chromatographies or ELISA is a widely distributed phenomenon known as Abs complexation polyspecificity or polyreactivity [64,65,66,67].

It was shown that, similar to normal enzymes, Abzs against many proteins usually split specifically only one specific protein–antigen and cannot hydrolyze many other control globular proteins ([19,20,21,22,23] and refs therein). It was shown that anti-MBP Abzs of patients with several ADs could hydrolyze only MBP [36,37,38,39,40], while abzymes against histones—only histones [44,45,46,47,48,49]. Catalytic cross-activity of any Abzs against different proteins has not yet been discovered [19,20,21,22,23]. However, recently, it was first demonstrated that IgGs of HIV-infected patients against MBP split specifically MBP and five H1-H4 histones and vice versa—abzymes against histones effectively hydrolyze MBP [48,49]. Production of such abzymes with cross-catalytic reactivity could be hazardous for developing many ADs since Abs against histones can hydrolyze MBP of nerve tissue cells. It seemed interesting to examine to what extent the phenomenon of enzymatic cross-reactivity between abzymes against MBP and histones is common for humans and animals with different ADs. 

As indicated above, abzymes, hydrolyzing MBP and histones are produced in the blood of C57BL/6 mice during the development of EAE. However, it remained unclear whether antibodies against histones and MBP possess polyspecificity in recognition and catalytic cross-reactivity. In this work, the analysis of the ability of C57BL/6 mice antibodies against H4, H1, H2A, H2B, and H3 histones and MBP to hydrolyze H4 histone was undertaken for the first time. It was shown that abzymes of mice against five histones and MBP possess polyreactivity in complex formation and unusual catalytic cross-reactivity in the hydrolysis of H4 histone. Moreover, it has been demonstrated that abzymes against five histones from the blood plasma of mice at different stages of EAE development can hydrolyze H4 with different efficiency and in various specific sites.

## 2. Results

### 2.1. Choosing a Model for the Study of Catalytic Cross-Reactivity

The development of EAE in prone C57BL/6 mice occurs spontaneously. Immunization of mice with DNA–protein complexes [11,12] and myelin oligodendrocyte glycoprotein (MOG) accelerates the development of EAE [13,14]. There are several stages of the EAE development: the onset on days 7–8, the acute phase on days 18–20, and the remission stage 25–30 days after immunization. The acceleration of EAE development is associated with specific changes in bone marrow HSCs differentiation profiles, an increase in lymphocyte proliferation, and repression of apoptosis in different organs of mice [11,12,13,14]. These processes are bound in parallel with the production of Abzs splitting DNA, MBP, MOG, and histones. The parameters characterizing all these changes were investigated earlier in [11,12,13,14]. To study the enzymatic cross-reactivity of IgGs, we chose two models; mice immunized with MOG [13,14] and with DNA–histones complex [11]. Data on changes in the differentiation profile of stem cells before and after immunization with MOG and DNA–histones complex are provided in the Appendix A. Data on the change in the relative concentrations of antibodies against DNA, MOG, and histones during the development of EAE are presented in Appendix A. The changes in the relative activities of IgGs in the hydrolysis of DNA, MOG, MBP, and histones during the development of EAE are shown in Appendix A. It can be seen that during spontaneous in time development of EAE, the increase in the relative amounts of all four precursors of hemopoietic cells (BFU-E, CFU-E, CFU-GM, and CFU-GEMM) in the bone marrow of the mice is relatively slow and gradual. Treatment of mice with MOG and DNA–histones complex results in different changes in the profile of stem cell differentiation over time. However, in all cases, EAE development accelerates.

Here, we study possible enzymatic cross-reactivity IgGs of C57BL/6 mice against five histones and MBP, taking into account the previously obtained data on the change in the activity of Abs-abzymes from the blood of C57BL/6 mice before and after their immunization with various antigens [11,12,13,14,68]. It was shown that the acceleration of the development of EAE after immunization of mice with MOG and complex of DNA with histones differs to some extent. Immunization of mice with MOG leads to the onset of the pathology by 7–8 days (the appearance of abzymes) and a sharp exacerbation in the acute phase at 17–20 days (maximum activity of abzymes) followed by a slow transition to the stage of remission and a decrease in the activity of abzymes in the hydrolysis of DNA, MBP, MOG, and histones. After immunization of mice with a complex of DNA–histones, the first peak of activation of EAE development is observed in 7–20 days. Still, the activity of abzymes increases more strongly in the period from 30 to 60 days. Therefore, the following groups of mice were used for analysis of H4 histone hydrolysis sites with IgGs against histones and MBP corresponding to different stages of EAE.

Con-0d: two types—against histones (Con-0d-His) and MBP (Con-0d-MBP), non-immunized control mice, blood sampling was carried out on the day of the beginning of the experiment.Con-60d: two types—against histones (Con-60d-His) and MBP (Con-60d-MBP), non-immunized control mice corresponding to spontaneous development of EAE, blood sampling was performed 60 days after the start of the experiment.MOG-20d: two types—against histones (MOG-20d-His) and mbp (MOG-20d-MBP), mice immunized with Mog, blood sampling was performed 20 days after the immunization of mice.DNA-20d: one type—against histones (DNA-20d-His), mice immunized with a complex of DNA and histones, blood sampling was carried out 20 days after the immunization of mice.DNA-60d: two types—against histones (DNA-60d-His) and MBP (DNA-60d-MBP), mice immunized with a complex of DNA and histones, blood sampling was carried out 60 days after the immunization of mice.

IgG antibodies against MBP and five histones were isolated from the blood of mice of these groups, and their relative activity in the hydrolysis of H4 histone was performed.

### 2.2. Purification of Antibodies

To analyze the “average” site-specific cleavage of H4 by IgGs against five histones and MBP, we first obtained electrophoretically homogeneous IgG preparations (IgG_mix_) from the mixture of seven plasma blood samples corresponding to each of five groups of mice mentioned above. The blood plasma proteins were first isolated on Protein G-Sepharose in conditions allowing the removal of nonspecifically bound proteins [34,35,36,37,38]. Then, IgG_mix_ preparations were additionally purified using FPLC gel filtration under drastic conditions (pH 2.6), destroying immune complexes as in [36,37,38,39,40,41,42,43,44,45,46,47,48,49]. After SDS-PAGE of the mixture of IgG_mix_ preparations, proteolytic activities in the hydrolysis of histones and MBP were revealed only in one IgG protein band (Figure 1).

The relative activities were isolated from every IgG_mix_ preparation (mixture of seven IgGs against five histones corresponding to each of the mouse groups) by chromatography on histone5H-Sepharose. The IgG fraction non-specifically bound and with low affinity to five histones was first eluted with 0.2 M NaCl. Specific anti-histones IgGs having a high affinity for histone5H-Sepharose were eluted with Tris-Gly buffer, pH 2.6. For additional purification of anti-histones IgGs from potential impurities of Abs against MBP, the fraction from histone5H-Sepharose was passed through MBP-Sepharose. The fraction obtained at loading onto MBP-Sepharose was further used as anti-histone IgGs.

The IgG_mix_ fraction eluted from histone5H-Sepharose at loading was used to obtain anti-MBP IgGs using affinity chromatography on MBP-Sepharose. IgGs with low affinity for MBP were eluted using a buffer containing NaCl (0.2 M). Finally, anti-MBP IgGs were eluted using an acidic buffer (pH 2.6). For additional purifications of anti-MBP IgGs against potential impurities of anti-histones IgGs, the fractions eluted from MBP-Sepharose were subjected to re-chromatography on histone5H-Sepharose. The fraction of IgGs eluted from the sorbent at the loading was named anti-MBP IgGs. 

### 2.3. SDS-PAGE Analysis of Histones and MBP Hydrolysis 

Polyclonal unseparated IgGs from the blood sera of HIV-infected patients as shown earlier [44,45,46,47,48,49] and patients with MS effectively split both five human histones [40] and MBP [36,37,38,39]. Moreover, IgGs of HIV-infected patients against MBP and five histones possess polyspecific complex formation and enzymatic cross-reactivity in the hydrolysis of five histones and MBP [48,49]. It was interesting whether IgGs of EAE mice against five histones can also split both five histones and MBP and vice versa. To analyze a possible enzymatic cross-reactivity, we first used the fraction of anti-histone IgGs (eluted from histone5-Sepharose) and anti-MBP IgGs (eluted from MBP-Sepharose). Figure 2A,B demonstrates hydrolysis of H4 histone by IgGs against five histones and anti-MBP IgGs, while Figure 2C,D shows hydrolysis of MBP by these IgGs. 

The efficiency of H4 and MBP hydrolysis with different IgGs was judged from the decrease in these proteins in the initial bands after incubation with antibodies compared to their content in control—incubation without antibodies (Figure 2). After 12 h of the incubation with IgGs against histones and MBP, the relative content of H4 and 18.5 kDa MBP form decreased remarkably compared to the control experiment (lanes C). 

These data may potentially point out that anti-MBP and anti-histones IgGs of C57BL/6 mice could exhibit non-specific complex formation polyreactivity [64,65,66,67] and enzymatic cross-reactivity in MBP and histone hydrolysis. These findings, however, cannot provide truthful evidence of enzymatic cross-reactivity between IgG-abzymes against five histones and MBP, because, even after their isolation using several affinity chromatographies, one cannot exclude the possibility that recovered antibodies contain very small admixtures of alternative IgGs. The best proof of enzymatic cross-reactivity may be achieved from an undeniable difference in the specific sites of the histones hydrolysis by IgGs against MBP and histones. This study first analyzed the possibility of hydrolysis of histone H4 with specific IgG-abzymes against five histones and MBP.

### 2.4. MALDI Analysis of H4 Histone Hydrolysis 

As shown by the example of abzyme antibodies from the blood of HIV-infected patients with IgGs against five histones, they hydrolyzed all histones and MBP and vice versa [42,43,44,45,46]. In addition, it was shown that during the development of EAE in C57BL/6 mice, after their immunization with MOG or DNA–histones complex, three stages can be distinguished: onset (7–8 days), acute (17–20 days), and remission (>30 days) phases [8,9,11,12,13,14]. The development of EAE is associated with the production of auto-Abs against DNA, MOG, MBP, and histones [11,12,13,14]. Moreover, these autoantibodies can be without or abzymes with catalytic activities [11,12,13,14,68]. 

From a theoretical point of view, the immune system of humans and animals can develop up to a million various antibodies against the same antigen, which may differ in their very different characteristics [18]. Using monoclonal antibodies, it was demonstrated that the total pool of antibodies to each of these antigens could contain from 30 to 40% of antibodies-abzymes possessing different enzymatic activities [21,22,23]. These abzymes differ in their affinity for cognate substrates, optimal pH values, dependence, and independence from ions of one and two valence metals ions, etc. 

The analysis of the relative content of antibodies without catalytic activity and abzymes hydrolyzing DNA, MOG, MBP, and histones was performed for the first time in [68]. It was shown that each of the three stages of EAE development is characterized by a specific ratio of IgGs without catalytic activity and abzymes. Moreover, the ratio of antibodies without activity and those hydrolyzing these substrates at each stage significantly depends on the antigen used [68]. At each stage of EAE development, the main antigens against which abzymes are produced can be in a free state or be associated with blood component specific characteristic of this stage. Each of the antigens analyzed by us has several antigenic determinants against which antibodies may be produced. For example, MBP has four antigenic determinants (AGDs) [69], while different histones from 2 to 4 [70,71], while H4—2 AGDs [72,73]. In principle, various AGDs located on the surface of protein molecules can form complexes with other different molecules: peptides, proteins, oligosaccharides, lipids, and nucleic acids. Considering these factors, it could be hypothesized that formation of autoantibodies against each of the histones may depend on the antigen and stage of EAE development, thereby leading to the production of abzymes differing in the sites of their hydrolysis. In this work, for the first time, using the example of abzymes hydrolyzing histone H4, an analysis of the hydrolysis sites of this histone was carried out depending on the antigen (MOG or DNA–histone complex) and the stage of EAE development.

The IgG fractions having a high affinity to histones and MBP were used to reveal the cleavage sites of H4 by MALDI TOFF mass spectrometry. Incubation of IgGs in the absence of H4 histone did not lead to the appearance of detectable peaks in the region from 3 to 15 kDa (Figure 3A). Right after the addition of the IgGs (Figure 3B), H4 histone was almost homogeneous, demonstrating two signals corresponding to one- (*m*/*z* = 11230.3 Da) and two-charged ions (*m*/*z* = 5616.7 Da).

First, H4 cleavage analysis was carried out with IgGs against histones. The sites of hydrolysis and the efficiency of H4 cleavage by each of the IgG preparations used were established based on an average of 7–10 independent spectra. After 6 h of incubation of H4 with IgGs against five histones, only six reliably detectable peaks of its hydrolysis with molecular masses (MMs) > 5.6 kDa revealed (Figure 3B). The spontaneous development of EAE in mice during 60 days led to the appearance of more active abzymes that hydrolyze H4 at 12 sites (Figure 3C). Interestingly, the number of H4 hydrolysis sites by anti-histone antibodies 20 days after immunization of mice with a DNA–histone complex (DNA-20d-His) was found to be 9 (Figure 4A). 

Sixty days after immunization of mice with the DNA–histone complex (DNA-60-His), nine peaks were found corresponding to the H4 hydrolysis sites by anti-histones abzymes (Figure 4B). However, only six peaks corresponded to different sites of H4 abzyme hydrolysis with DNA-60d-His are the same as those for DNA-20d-His (Figure 4A,B). DNA-20d-His abzymes did not split H4 at three sites in the N-terminal region of H4 compared to Con-60d-His Abs (Figure 4B). Anti-histone abzymes, following mouse immunization with MOG (MOG-20d-His), hydrolyzed H4 histone in eight sites, which partially coincided with those for other anti-histone IgGs (Figure 4C). These data indicate that at time zero of the experiment, the blood of three-month-old mice contains a smaller number of abzymes capable of hydrolyzing histone H4 than after spontaneous development of EAE during 60 days. Moreover, during the spontaneous development of EAE or after immunization of mice using a complex of DNA with histones or MOG, the number of H4 hydrolysis sites can be changed, and abzymes can hydrolyze H4 histone at other sites.

An interesting question was also about the existence of catalytic cross-reactivity in the case of abzymes against histones and MBP and about the similarity and difference in the H4 hydrolysis sites by antibodies against these proteins. Similar to antibodies against MBP from the blood of HIV-infected patients [48,49], anti-MBP abzymes of C57BL/6 mice efficiently hydrolyzed histone H4. Figure 5A shows typical MALDI mass spectra of H4 histone hydrolysis by anti-MBP IgGs corresponding to zero time (Con-0d-MBP preparation), demonstrating six peaks of histone hydrolysis. After spontaneous development of EAE for 60 days, as in the case of Abs against histones (Con-60d-His), an increase in the number of peaks corresponding to hydrolysis at 12 sites is observed for anti-MBP IgGs (Con-60d-MBP) (Figure 5B). 

Immunization of mice with MOG leads to the production of abzymes (MOG-20d-MBP) that cleave H4 at 13 sites (Figure 5D). Somewhat surprisingly, 60 days after immunization with the DNA–histone complex, the number of reliably detected peaks in the case of DNA-60d-MBP is only six (Figure 5C). Additionally, only two of them coincide with those for Con-0d-MBP (Figure 5). 

All H4 histone hydrolysis sites by anti-histone antibodies are summarized in Figure 6. 

It can be seen that, overall, the sites of histone hydrolysis by IgGs corresponding to the spontaneous development of EAE and different stages of accelerated development after immunization of mice with a complex of DNA with histones and MOG are substantially different and are predominantly located in clusters of different lengths.

Data on the sites of H4 splitting by different IgGs against MBP are shown in Figure 7. 

One could see that at the beginning of the experiment (time zero; Figure 7A), the number of H4 hydrolysis sites by anti-MBP IgGs (Con-0d-MBP) is only 6, and there is no protein splitting in its N-terminal zone (1–38 amino acid (AAs) residues). Hydrolysis sites in zone 1–34 AAs of H4 by anti-MBP IgGs appear after 60 days of spontaneous development of EAE (Con-60d-MBP; Figure 7B). Interestingly, there are significantly more cleavage sites after spontaneous development of EAE (12 sites; Figure 7B; Con-60d-MBP) and immunization of mice with MOG (11 sites; MOG-20d-MBP; Figure 7D). In this case, the main sites of H4 hydrolysis in both cases correspond to clusters located in the N-terminal zone of the protein—1–38 AAs (Figure 7B,D). It should be noted that all used preparations of IgGs against MBP hydrolyze H4 histone predominantly in different sites. 

For a more straightforward analysis of the coinciding and different sites of hydrolysis of H4 with IgGs against histones and MBP, they are presented in Table 1. 

It can be seen that 10 hydrolysis sites occur only once (marked with italics), and they are specific characteristics of some IgG preparations. At the same time, some hydrolysis sites are common from two to five preparations of IgGs. Only two close sites of H4 hydrolysis (R39-R40 and R40-G41) are common for six of the nine IgG preparations (Table 1). However, these, like other sites, in the case of different IgGs, differ being major, moderate, and minor ones.

## 3. Discussion 

The complexation polyreactivity of different antibodies is a widespread phenomenon [64,65,66,67]. Abs affinity for unspecific molecules is usually significantly lower than for cognate antigens, and they can be usually removed at affinity chromatography by 0.1–0.15 M NaCl [19,20,21,22,23]. Therefore, we eluted nonspecifically bound IgGs using 0.2 M NaCl. IgGs against five histones and MBP were additionally passed through alternative affinity sorbents. Finally, IgG fractions against MBP and five histones containing no alternative IgGs were obtained. It was shown that IgGs of EAE mice used in this study do not contain any normal proteases (Figure 1). The comparison of H4 hydrolysis sites with IgGs against MBP and five histones confirms this conclusion well. Trypsin splits various proteins after lysine (K) and arginine (R) residues. The H4 sequence contains 25 sites for potential cleavage of H4 by trypsin. However, the number of sites of H4 cleavage by all IgGs used after K and R residues varies mainly from 2 to 7 (Figure 6 and Figure 7). Chymotrypsin hydrolyzes proteins after F, Y, and W aromatic AAs. There are six potential sites of H4 histone hydrolysis by chymotrypsin. Only one site of hydrolysis after F (61F-62L) was found in the case of three out of nine used IgG preparations. Not a single site of H4 hydrolysis was found after the Y residue (Figure 6 and Figure 7, Table 1).

Interestingly, the sites of hydrolysis are mainly grouped in specific clusters of the histone H4 sequence. In addition, sites of splitting more often occur after neutral AAs: G, A, L, Q, T, and V (Figure 6 and Figure 7, Table 1). The sites of H4 specific hydrolysis by nine IgG preparations do not correspond to those for trypsin and chymotrypsin. They are not distributed along the entire protein length but are grouped into particular AA clusters. 

The primary evidence that the IgGs against five histones and MBP do not contain at least noticeable impurities of alternative abzymes follows the mismatch of the H4 hydrolysis sites by Abs against histones and MBP (Figure 6 and Figure 7, Table 1). This indicates that IgGs against histone H4 and MBP have not only polyspecificity of complexation but also possess enzymatic cross-reactivity. In addition, these data, together with previously published results, indicate that the phenomenon of polyspecificity of complex formation and enzymatic cross-reactivity are characteristic of abzymes against histones and MBP, not only for IgGs from the blood of HIV-infected patients [48,49] but also for experimental mice predisposed to EAE. We have previously shown that in the case of each immunogen (MOG and complex DNA–histones) at different stages of EAE development, remarkably or significant differences are observed in the differentiation profiles of bone marrow stem cells, level of lymphocyte proliferation in different organs, and in the relative activity of abzymes in the hydrolysis of DNA, MBP, MOG, and histones [11,12,13,14,68]. Despite significant differences, the immunization of mice predisposed to EAE with different antigens ultimately accelerates the development of this pathology. 

As previously shown, IgG antibodies against five different histones hydrolyze specific histones mainly at sites corresponding to their antigenic determinants [45,46,47,48,49]. However, different histones have antigenic determinants characterized by a high level of homology of protein sequences. This may be the main reason for hydrolysis of H4 by Abs against H4 histone and MBP. Using IgGs from the blood of HIV-infected patients against MBP and individual Abs against each of the five histones, it was shown that the main reason for catalytic cross-reactivity might be a consequence of the high level of protein sequence homology between MBP and histones [48,49]. Therefore, it was helpful to estimate the general homology between the protein sequence of H4 with four other histones and MBP. 

Complete identity of AAs between H4 and H1 (three alignment) was from 25.2 to 30.5% (average value 27.5 ± 2.7%), while similarity (identical together with non-identical amino acids but with highly similar physicochemical properties) varied from 48.6 to 58.4% (average 54.4 ± 5.1%). Identity of AAs between H4 and H2A varies from 28.2 to 32.8% (average value 30.5 ± 3.3%) and similarity from 49.1 to 54.6% (average 51.4 ± 4.6%). Approximately the same homology demonstrated H4 and H2B histones: identity—24.4–25.4% (average 24.8 ± 0.5%); similarity—49.0–54.1% (average value 51.0 ± 3.0%). Homology between H4 and H3 histones: identity—30.3%; and similarity—52.3% (there was only one variant of alignment found).

Since antibodies against MBP efficiently hydrolyze H4 histone, it seemed interesting to assess the homology of the MBP sequence with H4 and other histones. H4 histone has identity of AAs with MBP 25.0–29.4% (average 27.2 ± 3.1), while similarity is 46.2–48.6 (average 47.4 ± 1.1%). The average level of MBP identity with four other histones varied from 24.4 to 26.9% (average 25.7 ± 1.0%), while similarity from 45.5 to 50.8% (average value 48.6 ± 2.2%). The indicators of the AA identity of the sequences of all five histones (24.8–30.3%) and MBP (25.7–27.2%) and similarity in the case of five histones (51.0–54.4) as well as histones and MBP (46.2–50.8%) are very similar. In addition, all five histones and MBPs contain a large number of positively charged amino acid residues. Interestingly, not only five histones but also MBP effectively bind to DNA [73]. This may be several reasons for the possibility of antibodies against different histones and MBP binding with H4 and then hydrolysis of this histone (Table 1). However, as shown in [48,49] on the example of antibodies from the blood of HIV-infected patients, it is not the general level of homology between the complete sequences of histones and MBP. Still, the higher homology between the sequences that hydrolyze abzymes against MBP and histones in their cognate proteins is more important for abzyme enzymatic cross-reactivity.

In this work, using the EAE mice, we analyzed for the first time the possibility of changing the substrate specificity and sites of protein hydrolysis, depending on the stage of development of EAE. As shown in Figure 6A and Figure 7A and Table 1, the blood of three-month-old mice contains abzymes against histones and MBP, which hydrolyze histone H4 in both cases at six sites in total; only one of them (R55-G56) is the same for Con-0d-His and Con-0d-MBP. The spontaneous development of EAE within 60 days leads to a substantial increase in the number of H4 hydrolysis sites by both anti-histone (11 sites; Con-60d-His) and MBP (12 sites; Con-60d-MBP) antibodies. Moreover, for IgGs against histones at the beginning (Con-0d-His) and after 60 days of spontaneous development of EAE (Con-60d-His), there is a coincidence of only four sites of H4 cleavage. In the case of IgGs against MBP at time zero (Con-0d-MBP) and after 60 days of spontaneous development of EAE (Con-60d-MBP), only three of the same hydrolysis sites are observed (Table 1). As shown earlier, a change in the differentiation profile of stem cells, leading to the appearance of lymphocytes synthesizing abzymes, occurs already at the level of the cerebrospinal fluid of MS patients’ spinal cords [41,42,43]. Abzymes that hydrolyze DNA, MBP, and polysaccharides from the cerebrospinal fluid are about 30–60 times more active than the same individuals’ blood [41,42,43]. The difference in sites of H4 cleavage indicates that during the process of spontaneous development of EAE, there is an expansion of B-lymphocytes synthesizing antibodies against histones and MBP possessing catalytic cross-reactivity.

As shown earlier, the maximum increase in the relative activity of abzymes in the hydrolysis of DNA, MOG, and MBP after mouse immunization with MOG occurs during the acute phase of pathology at 17–20 days [11,12,13,14]. It could be expected that a sharp increase in the relative catalytic activity of abzymes at this time may be associated not only with an increase in the blood in the relative content of abzymes but also with expanding the ability of Abs to hydrolyze proteins at various sites. Nevertheless, 20 days after immunization of mice with MOG, abzymes against histones (seven sites; MBP-20d-His) showed significantly fewer H4 hydrolysis sites than after spontaneous development of EAE (Con-60d-His). Only three sites for MBP-20d-His preparation coincided with those for Con-0d-His antibodies, and four sites were the same with DNA-20d-His IgGs (Table 1).

Immunization of mice with the DNA–histones complex also accelerates the development of EAE. However, in this case, there are two stages of an increase in the enzymatic activity of abzymes. In the period from 7 to 20 days, the first increase occurs, and after 30 days, the second more powerful increase in their activity occurs [11,12]. A total of 20 days after immunization with a complex of DNA with histones, IgGs from the blood of mice (DNA-20d-His) show nine hydrolysis sites, and four of them (including two major sites of the splitting—R39-R40 and N64-V65) are not among the sites of H4 hydrolysis by antibodies (Con-0d-His) corresponding to mice before immunization. Additionally, 60 days after immunization, IgGs against histones (DNA-60d-His) hydrolyze H4 histone also at nine sites (Table 1). Five of these sites do not coincide with those for Con-0d-His, and three are different from sites for DNA-20d-His preparation.

With the spontaneous development of EAE, an increase in the relative activity of abzymes occurs relatively slowly and smoothly in comparison with those after immunization of mice with MOG or a complex of DNA with histones [11,12,13,14]. Therefore, it is somewhat unexpected that it is during the spontaneous development of pathology that there may be a more intensive expansion in the number of abzymes with different properties, which hydrolyze H4 at a greater number and different sites. This may mean that immunization of mice with MOG and complex DNA–histones leads to the production of specific abzymes that hydrolyze H4 at a smaller number of sites.

At present, the question of why abzymes can differ significantly at different stages of EAE development has remained open. First, IgG might be maturated during immunization. At the same time, various types of IgG might be produced in parallel by different B-cells. In the latter case, the poly-reactive function of IgGs can be well and easily explained.

Theoretically, the immune system can provide up to 10^6^ Ab variants against one antigen [74]. We have analyzed many monoclonal antibodies against DNA and MBP of SLE patients [75,76,77,78,79]. It was shown that the possible number of abzymes with DNase and MBP hydrolyzing activities that differ in optimal pH, dependence and independence from monovalent and divalent metal ions, exhibiting the properties of different DNases (DNase I and DNase II) or proteases (serine- and thiol-like, or metalloprotease) may be ≥1000. These data on monoclonal Abs are more in favor of the fact that the formation of B-lymphocytes producing antibodies with different enzymatic properties can occur in parallel. However, it cannot be ruled out that some of the abzymes are formed as a result of their maturation during immunization.

## 4. Materials and Methods

### 4.1. Materials and Chemicals

All chemicals, an equimolar mixture of H1, H2A, H2B, H3, and H4 histones, and homogeneous individual H4 were from Sigma (St. Louis, MO, USA). Superdex 200 HR 10/30 and Protein G-Sepharose columns were from GE Healthcare (GE Healthcare, New York, NY, USA). MBP was obtained from the Molecular Diagnostics and Therapy Center (DBRC; Moscow, Russia). Sepharoses containing immobilized histones and MBP were prepared using the standard manufacturer’s protocol, BrCN-activated Sepharose (Sigma), MBP, or the mixture of five histones. Mouse oligopeptide MOG_35–55_ was from EZBiolab (Heidelberg, Germany). All preparations were free from any possible contaminants.

### 4.2. Experimental Animals 

Inbred C57BL/6 mice (3 months of age) were used recently to study possible mechanisms of spontaneous and antigen-induced EAE development [11,12]. They were obtained in a special mouse breeding facility of the Institute of Cytology and Genetics (ICG) using standard conditions free of any pathogens. All experiments with C57BL/6 mice were performed under protocols of the Bioethical Committee of the ICG (document number 134A of 7 September 2010), satisfying the humane principles for working with animals established by the European Communities Council Directive (86/609/CEE). The Bioethical Committee of the ICG supported this study. The relative overtime weight, titers of Abs against MBP and histones, the relative level of proteinuria (concentration of protein in the urine, mg/mL), and some other parameters characterizing EAE development were analyzed as in [11,12,13,14]. 

### 4.3. Antibody Purification

Electrophoretically homogeneous polyclonal IgGs from the blood plasma of mice were first isolated by affinity chromatography of plasma proteins on Protein G-Sepharose and then additionally by Fast protein liquid chromatography–gel filtration (FPLC) on a Superdex 200 HR 10/30 column [11,12,13,14]. For additional purification, central parts of IgG peaks after gel filtration and filtration through filters (pore size 0.1 µm) were used. 

Removal of all IgGs against five histones (H1, H2A, H2B, H3, and H4) from total preparation of polyclonal Abs was fulfilled using histone5H-Sepharose (5 mL) containing five immobilized histones. The column was equilibrated using 20 mM Tris-HCl, pH 7.5 (buffer A). After IgGs loading, the column was washed to zero optical density (A_280_) with buffer A. Adsorbed Abs with low affinity for five histones were first eluted using buffer A supplemented NaCl (0.2 M), and finally, anti-histones IgGs with high affinity for the histones were specifically desorbed using acidic buffer (0.1 M glycine–HCl, pH 2.6). The IgG fractions eluted from histone5H-Sepharose at loading and washing with 5 mL of buffer A were combined and used to obtain anti-MBP Abs using affinity chromatography on the MBP-Sepharose column (5 mL) equilibrated in buffer A. After the MBP-Sepharose washing to zero optical density (A_280_) with buffer A, adsorbed IgGs with low affinity for MBP were first eluted using buffer A and then this buffer containing NaCl (0.2 M). Finally, anti-MBP IgGs were eluted using acidic Tris-Gly buffer (pH = 2.6), similar to histone5H-Sepharose. 

For additional purifications of anti-histones IgGs from possible small hypothetical impurities of antibodies against MBP, the fractions eluted from histone5-Sepharose were subjected to re-chromatography on MBP-Sepharose. The fraction eluted at the loading was named anti-histones IgGs. The fractions of anti-MBP antibodies eluted with an acid buffer from MBP-Sepharose were re-chromatographed on the histone5-Sepharose to remove hypothetically possible admixtures of IgGs against histones. The fraction of IgGs eluted at the loading was named anti-MBP IgGs. 

To exclude possible traces of normal proteases, the IgGs against histones and MBP were subjected to the assay of MBP- and histone-hydrolyzing activities after their SDS-PAGE using eluates of gel fragments as in [11,12,13,14]. It was shown that only intact IgGs demonstrate protease activity, and no other protein bands or proteolytic activities in different fragments of gel were found. 

### 4.4. Proteolytic Activity Assay 

For analysis of protease activity of IgG-abzymes, the reaction mixtures (10–17 μL) contained 20 mM Tris-HCl (pH 7.5), 0.8–1.0 mg/mL MBP, or an equimolar mixture of five histones, or H4 histone, and 0.01–0.05 mg/mL IgGs against MBP or five histones according to [41,42,43,44,45,46]. All mixtures were incubated during 3–12 h at 37 °C. Then, the reactions were stopped by adding SDS to the final 0.1% concentration. The efficiency of hydrolysis of histones and MBP was analyzed using SDS-PAGE in 20% gel. All proteins were detected using silver or Coomassie Blue staining. All gels were scanned and then quantified using Image Quant v5.2 software (Media Cybernetics, LP, New York, NY, USA). The relative protease activities of antibodies were evaluated from the decrease in relative intensity of bands corresponding to initial non-hydrolyzed proteins after their incubation without IgGs compared with their content after incubation with different IgGs.

### 4.5. MALDI-TOF Analysis of Histones Hydrolysis 

H4 histone was hydrolyzed during 0–25 h using anti-MBP, anti-histones IgGs as described above. The aliquots of reaction mixtures (1–2 µL) were analyzed over time using MALDI mass spectrometry. The analysis of the H4 histone hydrolysis products was performed using the Reflex III system (Bruker Frankfurt, Germany): a 337 nm nitrogen laser VSL-337 ND, 3 ns pulse duration, sinapinic acid was used as the matrix. To 1.6 µL of the matrixes and 1.6 µL of 0.2% trifluoroacetic acid, 1.6 µL of the solutions containing hydrolyzed histone H4 were added, and 1–1.6 µL of the obtained mixtures were applied on the MALDI plates that were air-dried for the analysis. All MALDI spectra were calibrated using mixtures of oligopeptides and proteins standards II and I (Germany, Bruker Daltonic) in the internal and/or external calibration mode. The analysis of molecular masses and specific sites of H4 hydrolysis by different IgGs was performed using Protein Calculator v3.3 (Scripps Research Institute). 

### 4.6. Analysis of Protein Sequence Homology

The level of homology between peptides and proteins sequences was carried out using *lalign* (http://www.ch.embnet.org/software/LALIGN_form.html (accessed on 1 January 2008)).

### 4.7. Statistical Analysis

The results correspond to the average values (mean ± standard deviation) from 7–10 independent spectra for each preparation of IgGs against five histones and MBP.

## 5. Conclusions

Here, for the first time, we showed that IgG-abzymes from EAE-prone C57BL/6 mice against histones, and myelin basic protein (MBP) possess the ability to form complexes with H4 histone demonstrating polyreactivity in complexation and catalytic cross-reactivity hydrolysis of histone H4. It was shown that IgGs against histones and MBP at the beginning of experiment (3-month-old mice) and after spontaneous development of EAE differ in sites and their number in which hydrolysis occurs. Immunization of mice with MOG and a complex of DNA with histones leads to an acceleration of the development of EAE and an increase in the relative activity of antibodies in H4 hydrolysis. After 20 days of immunization, the appearance of antibodies is observed that hydrolyze H4 at other additional sites, the number of which increases moderately. The maximum number of different sites of H4 hydrolysis was found for antibodies against histones and MBP corresponding to 60 days after immunization of mice.

## Figures and Tables

**Figure 1 ijms-23-09182-f001:**
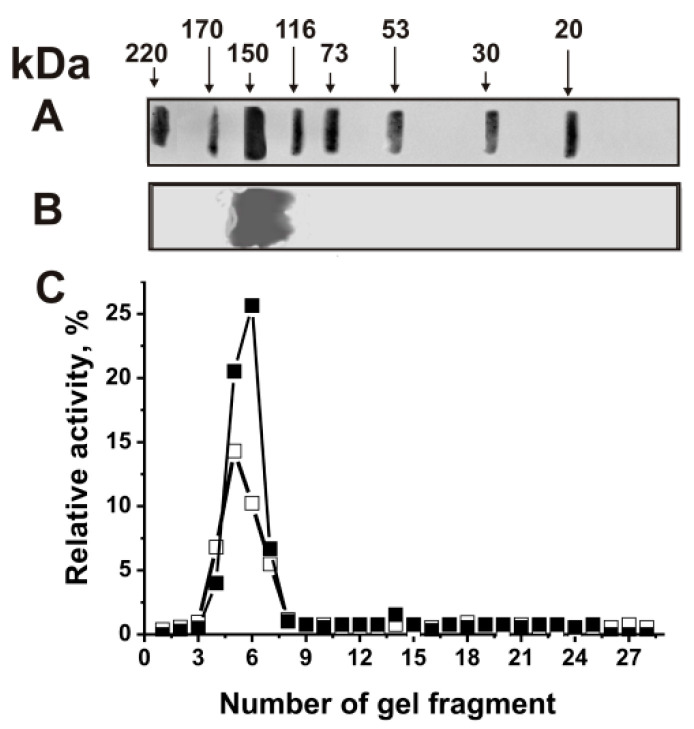
Panel (**A**) shows the position of molecular mass markers. The IgG_mix_ (14 μg) homogeneity analysis by SDS-PAGE under non-reducing conditions in the absence of thiol-disulfide reducing reagent dithiothreitol (**B**); silver staining. Panel (**B**) demonstrates the position of IgGs. The relative activities (RA, %) in the hydrolysis of five histones (■) and MBP (□) were estimated using eluates of gel fragments (2–3 mm) (**C**). After incubation for 24 h with eluates, complete hydrolysis of all substrates was undertaken for 100% (**C**). The errors of the relative activities estimation from two independent experiments did not exceed 7–10%.

**Figure 2 ijms-23-09182-f002:**
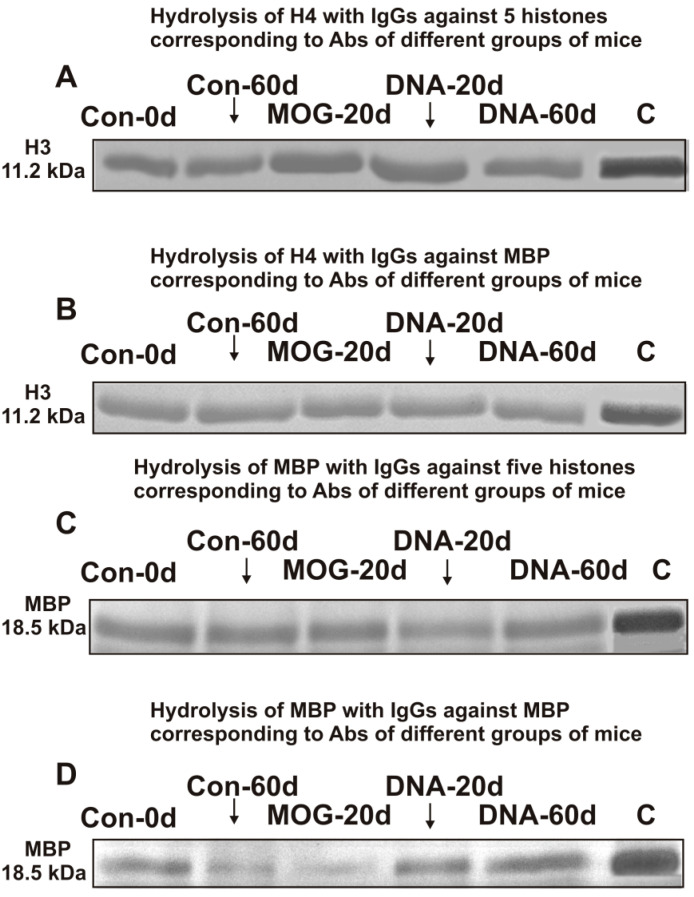
SDS-PAGE analysis of H4 histone hydrolysis by IgG-abzymes against five histones (**A**) and this histone with IgGs against MBP (**B**) as well as splitting myelin basic protein by IgGs against five histones (**C**) and IgG-abzymes against MBP (**D**). Lanes C correspond to the histones (**A**) and MBP (**B**) incubated without IgGs. MBP and a mixture of five histones with and without IgGs (0.03 mg/mL) were incubated for 12 h.

**Figure 3 ijms-23-09182-f003:**
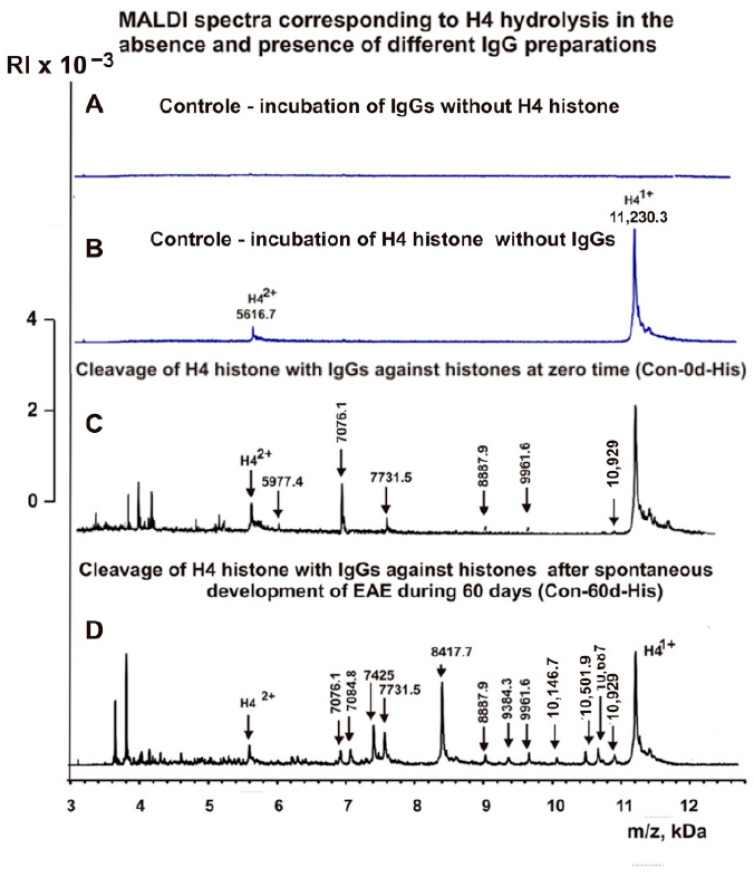
MALDI spectra corresponding to IgGs (0.045 mg/mL) incubated in the absence of H4 histone (**A**), products of H4 histone (0.8 mg/mL) hydrolysis in the absence (**B**) and the presence of IgGs ((**C**,**D**) 0.045 mg/mL) against five histones before immunization with antigens ((**C**) zero time—Con-0d-His preparation) and after spontaneous development of EAE during 60 days (Con-60d-His preparation) (**D**).

**Figure 4 ijms-23-09182-f004:**
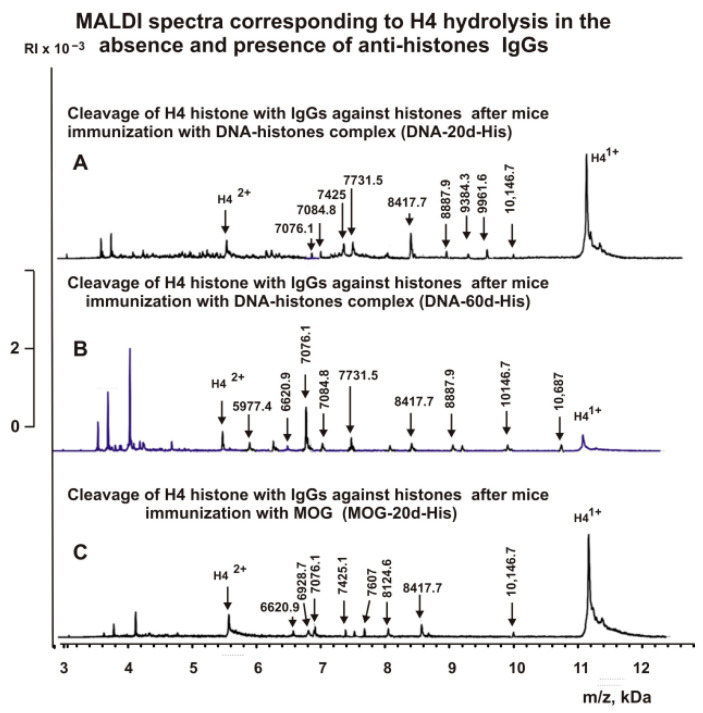
MALDI spectra corresponding to products of H4 histone (0.8 mg/mL) hydrolysis in the presence of IgGs ((**A**–**C**) 0.045 mg/mL) against five histones: DNA-20d-His ((**A**) IgGs against histones, 20 days after mouse immunization with a complex of DNA and histones), DNA-60d-His ((**B**) IgGs against histones, 60 days after mouse immunization with a complex of DNA and histones), and MOG-20d-His ((**C**) IgGs against histones, 20 days after mouse immunization with MOG).

**Figure 5 ijms-23-09182-f005:**
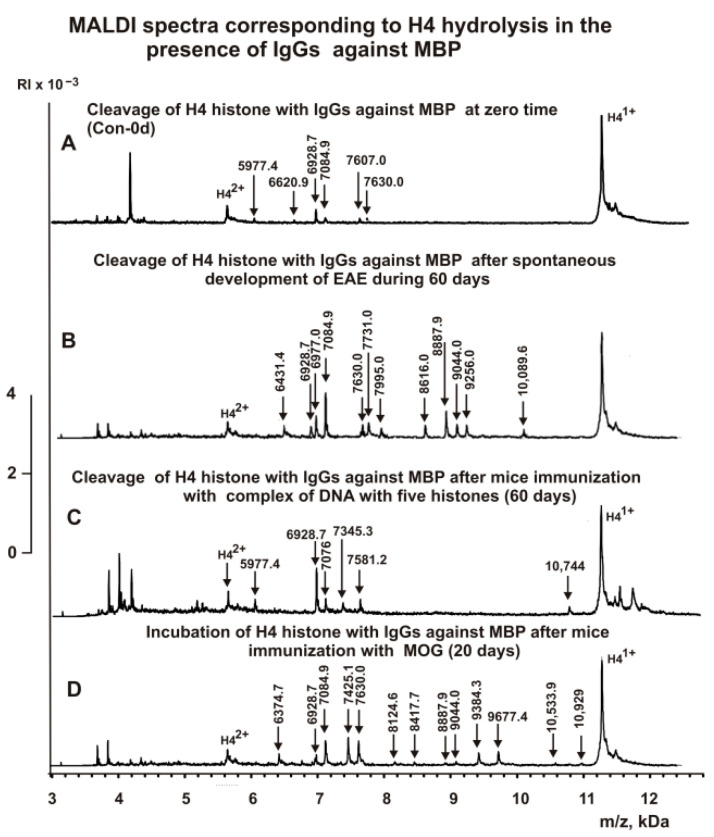
MALDI spectra corresponding to products of H4 histone (0.8 mg/mL) hydrolysis in the presence of IgGs against MBP ((**A**–**D**) 0.045 mg/mL) at zero time of the experiment ((**A**) Con-0d-MBP), after spontaneous development of EAE during 60 days ((**B**) Con-60d-MBP), 60 days after mouse immunization with DNA–histones complex ((**C**) DNA-60d-MBP), and 20 days after mouse treatment with MOG ((**D**) MOG-20d-MBP).

**Figure 6 ijms-23-09182-f006:**
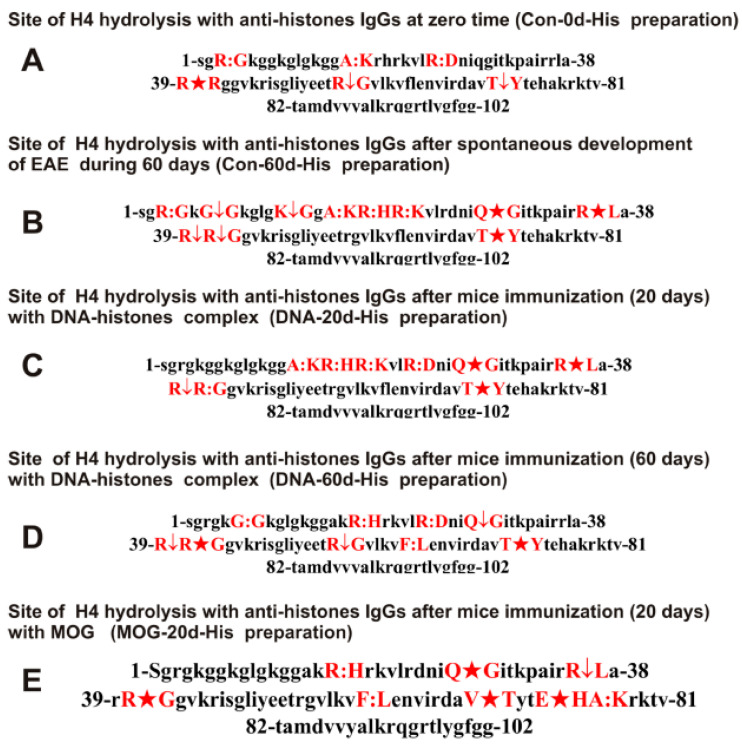
Sites of H4 hydrolysis by IgGs against histones corresponding to different IgG preparations. Major sites of H4 cleavage are shown by stars (★), moderate ones by arrows (↓), and minor sites of the cleavages by colons (:) (**A**–**E**).

**Figure 7 ijms-23-09182-f007:**
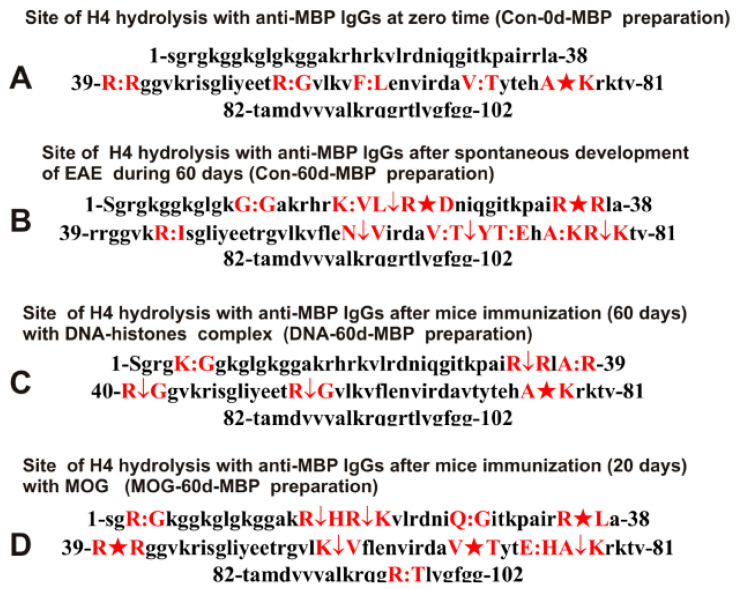
Sites of H4 hydrolysis by IgGs against MBP corresponding to different IgG preparations. Major sites of H4 cleavage are shown by stars (★), moderate ones by arrows (↓), and minor sites of the cleavages by colons (:) (**A**–**D**).

**Table 1 ijms-23-09182-t001:** Sites of H4 histone hydrolysis by IgGs against five histones and MBP *.

Molecular Mass Products of Hydrolysis (Da) and in the Brackets Corresponding to These Peptide Sites of H4 Hydrolysis								
Con-0d		Con-60d		DNA-20d	DNA-60d		MOG-20d	
Anti-His	Anti-MBP	Anti-His	Anti-MBP	Anti-His	Anti-His	Anti-MBP	Anti-His	Anti-MBP
10,929.0 (R3-G4)	-	10,929.0 ** (R3-G4)	-	-	-	-	-	10,929.0 (R3-G4)
-	-	10,687.0 (G6-G7)	-	-	10,687.0 (G6-G7)	-	-	-
-	-	-		-	-	* 10,744.0 * *** * (5K-6G) *	-	-
-	-	-	-	-	-	-	-	* 10,533.9 * * R95-T96 *
-	-	*10,501.9* *(K12-G13)*	-	-	-	-	-	-
-	-	10,146.7 (R17-H18)	-	10,146.7 (R17-H18)	10,146.7 (R17-H18)	-	10,146.7 (R17-H18)	-
-	-	-	* 10,089.6 * * (G13-G14) *	-	-	-	-	-
-	-	-	-	-	-	-	-	*9677.4* *(R17-H18)*
9961.6 (A15-K16)	-	9961.6 (A15-K16)	-	9961.6 (A15-K16)	-	-	-	
-	-	9384.3 (R19-K20)	-	9384.3 (R19-K20)	-	-	-	9384.3 (R19-K20)
			*9256.1* *(K20-V21)*					
-	-	-	9044.0 (L22-R23)	-	-	-	-	9044.0 (L22-R23)
8887.9 (R23-D24)	-	-	8887.9 ** (R23-D24)	8887.9 (R23-D24)	8887.9 (R23-D24)	-	-	8887.9 (R23-D24)
-	-	-	*8616.9* *(R78-K79)*	-	-	-	-	-
*-*	-	8417.7 (Q27-G28)	-	8417.7 (Q27-G28)	8417.7 (Q27-G28)	-	8417.7 (Q27-G28)	8417.7 (Q27-G28)
*-*	-	-	-	-	-	-	8124.6 (E74-H75)	8124.6 (E74-H75)
*-*	-	-	*7995.6* *(T73-E74)*	-	-	-	-	-
7731.5 (T71-Y72)	-	7731.5 (T71-Y72)	7731.5 (T71-Y72)	7731.5 (T71-Y72)	7731.5 (T71-Y72)	-	-	-
*-*	7630.0 (V70-T71)	-	7630.0 (V70-T71)	-	-	-	-	7630.0 (V70-T71)
*-*	*7581.2* *(R35-R36)*	-	-	-	-	*7581.2* *(R35-R36)*	-	-
*-*	-	7425.0 (R36-L37)	-	7425.0 (R36-L37)	-	-	7425.0 (R36-L37)	7425.0 (R36-L37)
*-*	-	-	-	-	-	*7345.3* *(A38-R49)*	-	-
*-*	7084.8 (R39-R40)	7084.8 (R39-R40)	7084.8 (R39-R40)	7084.8 (R39-R40)	7084.8 (R39-R40)	-	-	7084.8 (R39-R40)
7076.1 (R40-G41)	-	7076.1 (R40-G41)	-	7076.1 (R40-G41)	7076.1 (R40-G41)	7076.1 (R40-G41)	7076.1 (R40-G41)	
-	-	-	* 6977.0 * * (N64-V65) *	-	-	-	-	-
*-*	6928.7 (A76-K77)	-	6928.7 (A76-K77)	-	-	6928.7 (A76-K77)	6928.7 (A76-K77)	6928.7 (A76-K77)
*-*	6620.9 (61F-62L)	-	-	-	6620.9 (61F-62L)	-	6620.9 (61F-62L)	-
*-*	-	-	* 6431.4 * * (R45-I46) *	-	-	-	-	-
*-*	-	-	-	-	-	-	-	*6374.7* *(K59-V60)*
5977.4 (R55-G56)	5977.4 (R55-G56)	-	-	-	5977.4 (R55-G56)	5977.4 (R55-G56)	-	

* The molecular masses of the hydrolysis products and the corresponding hydrolysis sites were determined on the basis of a set of data from 7 to 10 spectra. ** Major hydrolysis sites are marked in red, moderate in black, and minor sites in blue. Missing hydrolysis sites are marked with a dash (-). *** Hydrolysis sites that are found only once in the case of any IgG are marked with an italic font.

## Data Availability

The data that supports the results of this study are included in the article and its Appendix A.

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
