# Peer review of "Experimental Autoimmune Encephalomyelitis of Mice: Enzymatic Cross Site-Specific Hydrolysis of H4 Histone by IgGs against Histones and Myelin Basic Protein"

_ijms, 2022, doi:10.3390/ijms23169182_

Round 1
Reviewer 1 Report
This paper describes about” Experimental autoimmune encephalomyelitis of mice: enzymatic cross site-specific hydrolysis of H4 histone by IgGs against histones and myelin basic protein”. As far as I know, this is the first paper for IgGs to acquire the catalytic function to cleave the antigen depending on the immunization period. This finding is important in the catalytic antibody field and will contribute to analyze the cause of autoimmune diseases in future. However, there are some questionable contexts and parts in the manuscript. Therefore, if the following few comments are satisfied, this paper is acceptable.
Followings are the comments for the author:
1, I Introduction is too long. Should be shortened, especially about the description for diseases. The explanations from line 24-48 at page 3 are better moving to discussion.
2, Regarding the description about catalytic antibodies in Introduction and Discussion, authors should cite more recent papers concerned with the advanced studies on catalytic antibodies, though many old references before 2010 were cited (see following paper).
1)S. A. Planque, Y. Nishiyama, S. Sonoda, Y. Lin, H. Taguchi, M. Hara, S. Kolodziej, Y. Mitsuda, V. Gonzalez, H. B. Sait, K.-i. Fukuchi, R. J. Massey, R. P. Friedland, B. O’Nuallain, E. M. Sigurdsson, S. Paul, Specific amyloid b clearance by a catalytic antibody construct. J. Biol. Chem. 290, 10229–10241 (2015).
2)Bowen, A., Wear, M.P., Cordero, R., Oscarson, S. & Casadevall, A. A monoclonal antibody to Cryptococcus neoformans glucuronoxylomannan manifests hydrolytic activity for both peptides and polysaccharides. J. Biol. Chem. 292, 417–434(2017).
3)Hifumi, E., Taguchi, H., Tsuda, H., Minagawa, T., Nonaka, T., Uda, T. A new algorithm to convert a normal antibody into the corresponding catalytic antibody. Science Advances, 6(13), eaay6441(2020).
4)Hifumi, E., Taguchi, H., Nonaka, T., Harada, T. & Uda, T. Finding and characterizing a catalytic antibody light chain, H34, capable of degrading the PD-1 molecule. RSC Chemical Biology, 2(13), January (2021).
3, For Panel A in Fig.1; the molecular size should be put along with the result of SDS-PAGE.
4, For Fig.2; comparisons of the shading by hydrolysis with the control are presented at around 11.2 or 18.5kDa for H3 and MBP, respectively. Authors should state about the fragmented bands of these, which mostly appeare at the SDS-PAGE, if the hydrolysis took place. Show the lower part of SDS-PAGE below 11.2 and 18.5kDa. The fragmented bands must be observed because MALDI spectra detected many fragmentations.
5, Authors are saying that the longer the immunization period become, the higher the catalytic function of IgGs became. However, in Fig. 2, the data about DNA-60d and DNA-20d seems against the phenomena. The longer immunization period gave more faded bands compared with that of the shorter one. It mayt be reversed in the figure, I guess.
6, For Fig.3-5. Many peaks are observed, maybe, coming from Histons. However, there is no data of IgGs as the control. Can you deny the scission of IgGs?
7, The authors’ discovery that IgGs acquire the catalytic function according with the period of immunization is certainly valuable. This fact should be revealed to the world. However, the reason why and how the events take place must be discussed. One IgG might be maturated during immunization, or various types of IgG might be produced. In the latter case, poly-reactive function of IgGs in this study can be well and easily explained. This analysis may lead to the generation of multiple cleavage site of abzymes.
8, Authors should discuss the catalytic sites of IgGs showing the hydrolysis in this study from the viewpoint of catalytic antibody.
9, If the authors perform the AA sequencing for the isolated or purified IgG, this paper will be much improved.
Author Response
This paper describes about” Experimental autoimmune encephalomyelitis of mice: enzymatic cross site-specific hydrolysis of H4 histone by IgGs against histones and myelin basic protein”. As far as I know, this is the first paper for IgGs to acquire the catalytic function to cleave the antigen depending on the immunization period. This finding is important in the catalytic antibody field and will contribute to analyze the cause of autoimmune diseases in future. However, there are some questionable contexts and parts in the manuscript. Therefore, if the following few comments are satisfied, this paper is acceptable.
Followings are the comments for the author:
1, I Introduction is too long. Should be shortened, especially about the description for diseases. The explanations from line 24-48 at page 3 are better moving to discussion.
ANSWER:
Sorry, the article describes a very unusual and unexpected situation of polyspecificity of antibodies in recognition and catalytic polyreactivity. Considering this, from our point of view, it was necessary to present data in the introduction that could help to understand the fundamental possibility of such a phenomenon. We showed our article to other immunologists who do not work with abzymes, with the question of what is superfluous or redundant in the introduction. They were told that they didn't see what needed to be removed, because it will be difficult to understand where new results comes from and how they are supported by preliminary data and results.
2, Regarding the description about catalytic antibodies in Introduction and Discussion, authors should cite more recent papers concerned with the advanced studies on catalytic antibodies, though many old references before 2010 were cited (see following paper).
1)S. A. Planque, Y. Nishiyama, S. Sonoda, Y. Lin, H. Taguchi, M. Hara, S. Kolodziej, Y. Mitsuda, V. Gonzalez, H. B. Sait, K.-i. Fukuchi, R. J. Massey, R. P. Friedland, B. O’Nuallain, E. M. Sigurdsson, S. Paul, Specific amyloid b clearance by a catalytic antibody construct. J. Biol. Chem. 290, 10229–10241 (2015).
2)Bowen, A., Wear, M.P., Cordero, R., Oscarson, S. & Casadevall, A. A monoclonal antibody to Cryptococcus neoformans glucuronoxylomannan manifests hydrolytic activity for both peptides and polysaccharides. J. Biol. Chem. 292, 417–434(2017).
3)Hifumi, E., Taguchi, H., Tsuda, H., Minagawa, T., Nonaka, T., Uda, T. A new algorithm to convert a normal antibody into the corresponding catalytic antibody. Science Advances, 6(13), eaay6441(2020).
4)Hifumi, E., Taguchi, H., Nonaka, T., Harada, T. & Uda, T. Finding and characterizing a catalytic antibody light chain, H34, capable of degrading the PD-1 molecule. RSC Chemical Biology, 2(13), January (2021).
ANSWER:
We have added these articles in the references
3, For Panel A in Fig.1; the molecular size should be put along with the result of SDS-PAGE.
ANSWER:
We have added positions of molecular mass marker on Fig. 1 (Lane A)
4, For Fig.2; comparisons of the shading by hydrolysis with the control are presented at around 11.2 or 18.5kDa for H3 and MBP, respectively. Authors should state about the fragmented bands of these, which mostly appeare at the SDS-PAGE, if the hydrolysis took place. Show the lower part of SDS-PAGE below 11.2 and 18.5kDa. The fragmented bands must be observed because MALDI spectra detected many fragmentations.
ANSWER:
Figure 2 shows a lot of data on protein hydrolysis by different antibody preparations. It is very difficult to give complete gel data for all preparations, since they will take at least three separate figures and it will be difficult to compare the relative histone hydrolysis by different IgG preparations if they are arranged in three separate figures. In addition, hydrolysis products stain poorly compared to the initial non-hydrolyzed protein. The efficiency of hydrolysis in all cases was assessed by the decrease in protein after hydrolysis in the initial histone band. It is not possible to estimate the molecular weights of the hydrolysis products from the SDA-PAGE data. Considering this, the estimation of molecular mass of the hydrolysis products was carried out using MALDI mass spectrometry.
5, Authors are saying that the longer the immunization period become, the higher the catalytic function of IgGs became. However, in Fig. 2, the data about DNA-60d and DNA-20d seems against the phenomena. The longer immunization period gave more faded bands compared with that of the shorter one. It may be reversed in the figure, I guess.
ANSWER:
This statement is not accurate. as shown in the works, the relative activity of antibodies in the hydrolysis of histones, MBP and DNA during the development of EAE is very dependent on the immunogen - MOG or the DNA-histone complex, as well as the hydrolyzed substrate: five histones, MBP and DNA. With this in mind, we have corrected this text and delete this statement,
6, For Fig.3-5. Many peaks are observed, maybe, coming from Histons. However, there is no data of IgGs as the control. Can you deny the scission of IgGs?
ANSWER:
When analyzing the histone hydrolysis spectra, we used the range from 2-3 to 15 kDa. In this range, antibody preparations do not show any peaks, since light, heavy chains and intact IgGs have a higher molecular weight. In the reaction mixtures 0.8 mg/ml H4, and only 0.045 mg/mL IgGs against MBP or five histones. Even when using the measurement range of 10-100 kDa, the peaks from IgGs at this concentration are very-very small (barely visible pimples). In addition, if there was a noticeable self-hydrolysis of IgGs and the peaks of hydrolysis products may be visible, then the molecular weights of these products would not correspond to the molecular weights of the histone hydrolysis products; they would be different and uninterpretable Taking this into account, we did not present the spectra of antibodies during their incubation without histone. However, taking your comments into account, we have shown a control spectrum corresponding to the incubation of antibodies in the absence of histone in Fig. 3A.
7, The authors’ discovery that IgGs acquire the catalytic function according with the period of immunization is certainly valuable. This fact should be revealed to the world. However, the reason why and how the events take place must be discussed. One IgG might be maturated during immunization, or various types of IgG might be produced. In the latter case, poly-reactive function of IgGs in this study can be well and easily explained. This analysis may lead to the generation of multiple cleavage site of abzymes.
ANSWER:
I'm sorry, but this question is still very complicated.
At present, the question of why abzymes can differ significantly at different stages of EAE development has remained open so far. First, IgG might be maturated during immunization. At the same time, various types of IgG might be produced in parallel by different B-cells. In the latter case, poly-reactive function of IgGs can be well and easily explained.
Theoretically, immune system can provide up to 106 Ab variants against one antigen [75]. We have analyzed many monoclonal antibodies against DNA and MBP of SLE patients [76-80]. It was shown that the possible number of abzymes with DNase and MBP hydrolyzing activities that differ in optimal pHs, dependence and independence from monovalent and divalent metal ions, exhibiting the properties of different DNases (DNase I and DNase II) or proteases (serine- and thiol-like, or metalloprotease) may be
³ 1000. These data on monoclonal Abs are more in favor of the fact that the formation of B-lymphocytes producing antibodies with different enzymatic properties can occur in parallel. However, it cannot be ruled out that some of the abzymes are formed as a result of their maturation during immunization.
8, Authors should discuss the catalytic sites of IgGs showing the hydrolysis in this study from the viewpoint of catalytic antibody.
ANSWER:
Sorry, but as mentioned above, the blood of SLE patients contains 1000 antibodies against one antigen with a variety of enzymatic properties. Approximately the same situation can be in the case of antibodies from the blood of sick mice. Given this, it is not yet clear how hydrolysis sites can be discussed in terms of multiple abzymes.
9, If the authors perform the AA sequencing for the isolated or purified IgG, this paper will be much improved.
ANSWER:
Sorry, but judging by the results of there are many abzymes with various catalytic activities and they remarkably differ in hydrolysis sites depending on the stage of EAE development. Given this, this is a big challenge that we plan to tackle in the future.
Thanks for the valuable comments
Best Wishes
Prof. Georgy Nevinsky
Reviewer 2 Report
The article by Andrey E. Urusov et. al. entitled " Experimental autoimmune encephalomyelitis of mice: enzymatic cross site-specific hydrolysis of H4 histone by IgGs against histones and myelin basic protein " is an interesting study on discovery of new type of catalytic antibodies having cross hydrolytic activity.
The authors immunized histones (H4, H1, H2A, H2B, and H3) and MBP to mice which is model of human multiple sclerosis. After several affinity purifications, they obtained IgG-abzymes with unusual poly reactivity in complexation and catalytic cross-reactivity in the hydrolysis of histone H4. Furthermore, they found that at different stages of experimental autoimmune encephalomyelitis development, abzymes could significantly differ in specific sites of H4 hydrolysis. In general, the article is written clearly and presents interesting data, even if the figures and tables are not well represented. This article would be of interest to scientists who are focusing on the study of catalytic antibody and autoimmune disease.
I have the following comments and concerns.
1. In Fig 2, is there any reason to measure hydrolysis activity of H4? The hydrolysis activity of other histones also should be measured and presented using same method. Hydrolysis activity of IgGs against MBP seems to be more than that against H4. Is there comment regarding these differences?
2. According to material and methods(P16), IgGs were purified from the blood of plasma. However, in the legend of Figure S3, IgGs were from serum. Is there any difference of hydrolysis activity of IgGs between the blood plasma and serum? Why is hydrolytic activity at 40 days lower than that at 20 days in Figure S3?
3. During antibody purification(P16-7), anti-MBP antibodies in anti-histone IgGs were removed using MBP-Sepharose and anti-histone antibodies in anti-MBP IgGs were removed using histone5-Sepharose. How do anti-histone IgGs recognize MBP molecule?
There are many careless grammatical and typographical errors throughout the manuscript. I would encourage the authors to proofread the manuscript carefully.
Author Response
The article by Andrey E. Urusov et. al. entitled " Experimental autoimmune encephalomyelitis of mice: enzymatic cross site-specific hydrolysis of H4 histone by IgGs against histones and myelin basic protein " is an interesting study on discovery of new type of catalytic antibodies having cross hydrolytic activity.
The authors immunized histones (H4, H1, H2A, H2B, and H3) and MBP to mice which is model of human multiple sclerosis. After several affinity purifications, they obtained IgG-abzymes with unusual poly reactivity in complexation and catalytic cross-reactivity in the hydrolysis of histone H4. Furthermore, they found that at different stages of experimental autoimmune encephalomyelitis development, abzymes could significantly differ in specific sites of H4 hydrolysis. In general, the article is written clearly and presents interesting data, even if the figures and tables are not well represented. This article would be of interest to scientists who are focusing on the study of catalytic antibody and autoimmune disease.
I have the following comments and concerns.
- In Fig 2, is there any reason to measure hydrolysis activity of H4? The hydrolysis activity of other histones also should be measured and presented using same method. Hydrolysis activity of IgGs against MBP seems to be more than that against H4. Is there comment regarding these differences?
ANSWER:
Sorry, but it is simply not possible to evaluate and describe hydrolysis of all the Histones in one article. It turns out that the data on the hydrolysis of each of the histones occupy the entire volume of one article. We are currently analyze the hydrolysis of other histones with antibodies against all histones under the same conditions and these data will be published later.
You are right, indeed, hydrolysis with antibodies against MBP is in some cases more effective than against H4 histone
- According to material and methods(P16), IgGs were purified from the blood of plasma. However, in the legend of Figure S3, IgGs were from serum. Is there any difference of hydrolysis activity of IgGs between the blood plasma and serum? Why is hydrolytic activity at 40 days lower than that at 20 days in Figure S3?
ANSWER:
Sorry in Fig. S3 there is an error - all antibodies were isolated from the plasma. This mistake has been fixed.
That the activity at 40 days is lower than at 20 days is a general trend in antibody activities after mice immunization of MOG. The maximum activity is always observed during the acute phase (17-20 days), and then during the remission period (more than 25-30 days) it always decreases [8,9,11-15,65].
- During antibody purification(P16-7), anti-MBP antibodies in anti-histone IgGs were removed using MBP-Sepharose and anti-histone antibodies in anti-MBP IgGs were removed using histone5-Sepharose. How do anti-histone IgGs recognize MBP molecule?
ANSWER:
As we have shown earlier in a number of works, that all histones are characterized by a high level of homology with the myelin basic protein.
Data from the text :
As previously shown, IgG antibodies against five different histones hydrolyze specific histones mainly at sites corresponding to their antigenic determinants [42-46]. But different histones have antigenic determinants characterized by a high level of homology of protein sequences. This may be the main reason for hydrolysis of H4 by Abs against H4 histone and MBP. Using IgGs from the blood of HIV-infected patients against MBP and individual Abs against each of five histones, it was shown that the main reason for catalytic cross-reactivity might be a consequence of the high level of protein sequences homology of MBP and histones [45-46].
And further in the article data on the level of homology between histones and MBP are given.
There are many careless grammatical and typographical errors throughout the manuscript. I would encourage the authors to proofread the manuscript carefully.
ANSWER:
We conducted additional analysis of the text and corrected errors.
Thanks for the valuable comments
Best Wishes
Prof. Georgy Nevinsky
Reviewer 3 Report
Review comments are attached

Author Response
The manuscript describes efforts to delineate the intricacies of autoimmune encephalomyelitis in mice. Specifically, due attention was provided and focus was directed toward delineation of profiles pertaining to the enzymatic cross site-specific hydrolysis of H4 histone by IgGs against histones and myelin basic protein (MBP). The work was conducted with due consideration of the fundamentals of the pathophysiologies investigated and the results obtained through chromatographic and spectroscopic/spectrometric techniques led to specifically formulated conclusions. However, there are a number of problems associated with the presentation of the work done in the manuscript. These concerns should be rectified by the authors prior to any consideration. A limited number of those problems is listed below: One serious impediment is language. The manuscript cannot be followed.
- In the Abstract, where the summarized work is to be presented by the authors to attract the attention of the reader, the use of acronyms without definition is overwhelming. Since the acronymic terms are used for the first time, they should be defined as well. That phenomenon has also been encountered for other acronyms in the text as well.
ANSWER:
It was done
- With the respect to the introduction, the text is overly stated, when it comes to defining the field of action by the authors. The length should be reduced and the introduction should be rewritten so that it becomes legible from start to finish. At the end of the introduction, the formulation of the work should be done in a way that it is comprehensible.
ANSWER:
The article describes a very unusual and unexpected situation of polyspecificity of antibodies in recognition and catalytic polyreactivity. Considering this, from our point of view, it was necessary to present data in the introduction that could help to understand the fundamental possibility of such a phenomenon. We showed our article to other immunologists who do not work with abzymes, with the question of what is superfluous or redundant in the introduction. They were told that they didn't see what needed to be removed. Given this, if it possible, please inform us more specifically what, in your opinion (what information) is redundant.
- Discontinuity in the terms of the various molecular targets involved in pathological processes creates disruption of the flow of thoughts that deters the reader from moving on with the actual work. That should change.
ANSWER:
Sorry, but we don't understand what you mean by “Discontinuity in the terms of the various molecular targets”. Therefore, it is not clear what and how the text can be changed
- Several paradigms of ill-written statements can be provided to project the problems in understanding for the contents of the manuscript. To that end, examples of such case are provided below:
- a) In the introduction section, the statement “Titers of auto-Abs to specific auto-antigens at the beginning of different ADs usually correspond to typical indices' ranges conforming to healthy humans and mice.” should be corrected to read “Titers of autoAbs to specific auto-antigens at the beginning of different ADs usually correspond to typical index ranges conforming to healthy humans and mice.”
Answer:
It was corrected
- b) In the Results section, the statement “There are several stages of the EAE development: the onset at 7-8, the acute phase at 18-20, and the remission stage later 25-30 days after immunization.” should be rewritten (I reckon) to read “There are several stages of the EAE development: the onset on day 7-8, the acute phase on day 18-20, and the remission stage 25-30 days after immunization.”
Answer:
It was corrected
- c) In the enumerated protocol experiments, the authors provide terms as such Con-0d, Con-60d. What do they mean by that? Do they mean “control”? The terms should be explicably stated prior to their use.
Answer:
Excuse me, but the part “Choosing a model for the study of catalytic cross-reactivity”
provides an explanation of these Con-0d, Con-60d designations for the types of antibodies used.
From text:
Con-0d: two types - against histones (Con-0d-His) and MBP (Con-0d-MBP), non-immunized control mice, blood sampling was carried out on the day of the beginning of the experiment;
Con-60d: two types - against histones (Con-60d-His) and MBP (Con-60d-MBP), non-immunized control mice corresponding to spontaneous development of EAE, blood sampling was performed 60 days after the start of the experiment;
- d) In section 2.3, the statement “These data may potentially point out that anti-MBP and anti-histones IgGs of C57BL/6 mice could possess a known phenomenon of unspecific complex formation polyreactivity [61-64] and enzymatic cross-reactivity in MBP- and histones hydrolysis.” should be corrected to read “These data may potentially point out that anti-MBP and anti-histones IgGs of C57BL/6 mice could exhibit non-specific complex formation polyreactivity [61-64] and enzymatic cross-reactivity in MBP and histone hydrolysis.”
Answer;
It was corrected
- e) In the same section, the statement “Considering these factors, it could assume that the formation of autoantibodies against each of the histones can depend on the antigen and the stage of EAE development leading to the production of abzymes differing in the sites of their hydrolysis.” should be corrected to read “Considering these factors, it could be hypothesized that formation of autoantibodies against each of the histones may depend on the antigen and stage of EAE development, thereby leading to the production of abzymes differing in the sites of their hydrolysis.”
Answer;
It was corrected
- f) In the same section, the statement “Anti-histone abzymes after mice immunization with MOG (MOG-20d-His) hydrolyzed H4 histone at eight sites partially coincided with those for other anti-histone IgGs (Figure 4C).” should be corrected to read “Antihistone abzymes, following mouse immunization with MOG (MOG-20d-His), hydrolyzed H4 histone in eight sites, which partially coincide with those for other antihistone IgGs (Figure 4C).”.
Answer;
It was corrected
- g) Toward the action of section 2.3, the statement “These findings, however, cannot provide truthful evidence of enzymatic cross-reactivity between IgG-ab-zymes against five histones and MBP because, even after their isolations using several affinity chromatographies, one cannot be excluded that obtained antibodies nevertheless could contain very small admixtures of alternative IgGs.” should be corrected to read “These findings, however, cannot provide truthful evidence of enzymatic cross-reactivity between IgG-abzymes against five histones and MBP, because, even after their isolation using several affinity chromatographies, one cannot exclude the possibility that recovered antibodies contain very small admixtures of alternative IgGs.”.
Answer;
It was corrected
- h) In the discussion section, the statement “… it was shown that the main reason for catalytic cross-reactivity might be a consequence of the high level of protein sequences homology of MBP and histones [45-46].” should be corrected to read “… it was shown that the main reason for catalytic cross-reactivity might be a consequence of the high level of protein sequence homology between MBP and histones [45-46].”. i) The interchangeable use of nouns and adjectives should be attended to and thus a number of binary terms should be corrected.
Answer;
It was corrected
For example: “histone-DNAs complexes” should be “histone-DNA complexes”, “Toll-like receptors activation” should be “Tolllike receptor activation”, etc. throughout the text.
Answer;
It was corrected
- In the paragraph concluding the Introduction section, the statement “Catalytic crossactivity of any Abzs against different proteins till yet was not discovered [19-23].” should be corrected to read “Catalytic cross-activity of any Abzs against different proteins has not yet been discovered [19-23].”
Answer;
It was corrected
- In the legend of Figure 1, the acronym DTT is used for the presence of reducing conditions. That should be explained. In a second point, there is no reason to use first the term “absence of non-reducing conditions” and then use the term DTT. Providing a simple articulation of the explanation is more straightforward.
Answer;
It was corrected
- In the legend of Figure 4, the lane C depiction demands explanatory information, as noted in the text.
Answer;
It was corrected
- In the explanatory notes of Table 1, it is stated that “Major hydrolysis sites are marked in red, moderate in black, and minor sites in green.” From the looks of the data in the table, there is only red, black and blue!! Corrections should be made.
Answer;
It was corrected as blue
- On a number of occasions the term canonical was used. In the context of this work it is my opinion that this “idealized state’ term for a biological molecule (proteases, enzymes, etc.) should be replaced by a more colloquial term, such as “normal” or “physiological” or any term befitting to the case examined.
Answer;
It was corrected for normal
Thank you very much for your very valuable and helpful comments.
Best wishes
Prof. Georgy Nevinsky